# Transient CFD Modelling of Air–Water Two-Phase Annular Flow Characteristics in a Small Horizontal Circular Pipe

## Jun Yao and Yufeng Yao *

School of Engineering, Frenchay Campus, University of the West of England, Coldharbour Lane, Bristol BS16 1QY, UK; jun.yao@uwe.ac.uk
* Correspondence: yufeng.yao@uwe.ac.uk; Tel.: +44-117-32-87084

**Abstract:** The liquid film formed around the inner walls of a small horizontal circular pipe often exhibits non-uniform distributions circumferentially, where the film is thinner at the top surface than the bottom one. Even with this known phenomenon, the problem remains a challenging task for Computational Fluid Dynamics (CFD) to predict the liquid film formation on the pipe walls, mainly due to inaccurate two-phase flow models that can induce an undesirable 'dry-out' phenomenon. Therefore, in this study, a user-defined function subroutine (ANNULAR-UDF) is developed and applied for CFD modelling of an 8.8 mm diameter horizontal pipe, in order to capture transient flow behaviour, flow pattern formation and evolving process and other characteristics in validation against experiments. It is found that CFD modelling is able to capture the liquid phase friction pressure drop about maximum of 30% in deviation, consistent to the correlated experimental data by applying an empirical correlation of Chisholm. Due to the gravity effect, the liquid film is generally thicker at the bottom wall than at the top wall and this trend can be further enhanced by increasing the superficial air–water velocity ratios. These findings could be valuable for HVAC industry applications, where some desirable annular flow features are necessary to retain to achieve high efficiency of heat transfer performance.

**Keywords:** annular two-phase flow; computational fluid dynamics; small horizontal circular pipe; user-defined function subroutine; liquid film thickness

## 1. Introduction

The air and water two-phase flow problem presents in many industrial applications such as steam generator, chemical processing and pipeline for transportation, etc. A wide range of flow regimes such as bubbly flow, slug flow, and annular flow can be found in various pipe systems of these industry devices [1,2]. Most recently, a renewable system such as the air and water heat pump appears to be a type of popular heating source that can be employed for both commercial and domestic applications due to the requirements of environment impact and the shortage of energy resources [3]. The state-of-the-art heat exchanger has played a key role in improving the operational efficiency of air and water heat pump systems through fluid evaporation and condensation processes, where the void fraction of gas and liquid often changes along the flow path, resulting in favourable two-phase flow patterns that can largely enhance thermal performance of the heat exchange device. The key factors that could affect this process include forces (i.e., inertia, buoyancy, surface tension and shear stress), closely coupled with two-phase flow characteristics and parameters (i.e., flow velocity, temperature, vapour quality and most importantly flow orientation) [1]. Thus, it is crucial to have a deep understanding of the correlations among these key parameters, with regard to the enhancement of two-phase flow performance, especially the annular flow in a small horizontal circular pipe. This will ensure the high efficiency of heat transfer performance in such a system, as demanded by industry.

The two-phase annular flow is characterised by the liquid film distributed around the pipe inner walls circumferentially, where the film thickness distribution is closely associated

with the kinematic performance of gas–liquid interfaces, i.e., the wave propagation along the flow moving direction and the gravitation effect. In a horizontal circular pipe, the gas (or vapour) phase tends to move upwards towards the centre of the pipe due to a high density ratio of air and water and the gravitation force effect that is perpendicular to the flow direction. Therefore, drainage is a common phenomenon that could promote annular flow behaviour in the pipe. An earlier observation of liquid film draining circumferentially, and the secondary flow influence was first made by Pletcher et al. [4] and later by Laurinat et al. [5] and Lin et al. [6] whilst measuring annular flow film thickness and distribution along a horizontal circular pipe. In parallel, Butterworth [7] and Jayanti et al. [8] proposed an illustration of asymmetry characteristics of liquid film thickness based on their investigations of the liquid wave propagation and finally the evolution to different two-phase flow patterns (see, e.g. Butterworth [7]; Jayanti et al. [8]), in alignment with a number of correlations proposed by other researchers. Among them, Schubring and Shedd [9–11] developed an empirical correlation to model the base film thickness using the gas dynamics kinetic energy theory to discriminate the wavy annular and the full annular flow structures, considering the film thickness ratio variation along the pipe circumferential direction. Their study concluded that the two-phase flow model has strong correlations with the pipe diameter and the gas superficial velocity (also see Schubring and Shedd [12]). Following these experimental works, William et al. [13] studied the evaluation of two-phase flow asymmetry over a wide range of low gas velocities, opposite to those symmetry flow patterns observed at high gas velocity. Donniacuo et al. [14] studied the eccentricity using a linear relation related to the position of the vapour core, based on the difference between the top and the bottom liquid film thicknesses associated with the flow parameters. All these earlier works have laid a solid foundation for further research.

The two-phase flow often exhibits strong instantaneous features primarily due to the instability mechanism [15]. The time-varying interface was often difficult to capture and it was not until the late 1960s when the development of analytical and computational technologies occurred (see, e.g., Ruspini et al. [16]; Boure and Mihaila [17]) that this was achieved. Whilst the thermal–hydraulic instability phenomena of flow patterns in a nuclear reactor have been studied extensively [18], its implication in other relevant industries, such as oil and chemical processes, refrigeration and renewable energy systems where the two-phase flow is predominant, is still not well understood, despite some basic research having been performed in the past decades. For example, the two-phase flow studies in a heat exchanger have been carried out by Ishii [19], Lahey and Podowski [20], Prasad et al. [21] and many others. Boure et al. [22] defined two-phase flow instabilities to be static and/or dynamic types at a macroscopic scale level, considering different dynamic effects, such as transient, inertia and compressibility, where the pressure (or flow rate) is used to depict the flow instability. Few simulation studies have explored the two-phase flow instability phenomena at a microscopic scale level, such as in a small-scale horizontal circular pipe, partly owing to very limited experimental and empirical data in the public domain to be used for the implementation of flow dynamics in simulation.

The complexity of two-phase flow is mainly the presence of multiple, deformable and moving interfaces where fluid properties will have sudden changes between different phases, resulting in multiscale (both spatial and temporal) flow structures. Despite the flow pattern being identifiable and recognisable using a flow pattern map (see, e.g., Rouhani et al. [23]; Dobson et al. [24]; Thome [25]), the equilibrium between different phases is largely dependent on a few key factors, such as (1) flow conditions (e.g., phase superficial velocity, flow pressure, flow temperature), (2) fluid properties (e.g., density, viscosity, surface tension) and (3) the orientation of configuration (e.g., the component design in alignment with the flow direction) [26]. In a horizontal pipe, the propagation of the liquid film along the pipe wall surfaces can be modelled by considering a multi-dimensional model of annular two-phase flow, based on the local liquid film flow rate. This modelling approach still requires the flow governing Equations to formulate several enclosure relationships to describe mass, momentum and energy transfers within the gas

phase and the liquid (film) phase, respectively, and any possible phase exchanges between the two phases. Therefore, the particular challenge for this type of problem is mainly the complexity of the dominant phenomena at the gas and the liquid interfaces, where the closure relationships are not yet well-established to properly resolve both the spatial and temporal scales of two-phase flow [27].

Recently, computational fluid dynamics (CFD) has been extensively developed to study two-phase annular flow [28]. From an industry application perspective, the CFD method is still computationally too expensive to accurately capture the gas and liquid interfaces, especially in annular flow. This is truly reflected by the modelling approach when the two-fluid model (e.g., the Eulerian–Eulerian framework) is used to treat the gas and the liquid dynamic motions. To improve the modelling efficiency, a two-dimensional liquid film model was proposed by Bai and Gosman [29] to couple the liquid film and the gas core flow into a unified framework for annular flow simulations (see, e.g. Adechy and Issa [30]; Meredith et al. [31]). This approach considered mass, momentum and energy transfers between the liquid film and the gas core flow. In principle, these enclosure relationships can be represented by applying either the Eulerian–Eulerian or the Eulerian–Lagrangian approaches, respectively.

In a two-phase flow simulation with the Eulerian–Eulerian approach, the gas–liquid mixture is assumed to be an interpenetrating continuum, based on the two-fluid two-phase flow model [32]. Alternatively, the liquid droplets can be tracked as a cloud of particles using the Eulerian–Lagrangian approach where the gas phase is still simulated as a single-phase continuum flow (see, e.g. Li and Anglart [28]; Caraghiaur and Anglart [33]). Although all two-phase flow regimes can be primarily modelled via specific codes, the computational time often takes too long in simulation, and this can grow exponentially for practical industry problems with complex flow regimes. This is mainly because the simulation needs to properly resolve the flow motions down to spatial and temporal resolved small scales, despite that some time saving can be made by using the Reynolds-averaged Navier–Stokes (RANS) method [2]. To address these critical issues, Westende van't et al. [34] made some efforts to study the influence of the secondary flow on the liquid film distributions on a horizontal pipe wall using the Eulerian–Lagrangian approach, assuming the droplets being dragged towards the flow path follow its counter-rotating elements so that droplets are transported to 'wet' the top wall of the pipe. Their model has been used to simulate the particle-laden turbulent flow in a pipe where droplets are treated as small solid spheres to encounter the wall roughness, inducing the secondary flow patterns. Unfortunately, the liquid film characteristics were not fully examined in their studies. Nevertheless, this type of modeling approach has important implications to facilitate further development of an efficient fluid system containing small diameter circular pipes, e.g., the heat exchanger, for which the ultimate goal is to carry out an optimal design at a relatively low cost and short turn-around computational time. It is the main motivation of this study.

The purpose of this research is to carry out transient CFD analysis of air–water two-phase annular flow to investigate the liquid film characteristics and its evolution in an 8.8 mm diameter horizontal pipe, i.e., a configuration experimentally studied by Schubring and Shedd [9–12]. By using experimental inflow conditions, two-dimensional (2D) liquid film CFD modelling will be performed at first to verify two user-defined function (UDF) subroutines (2DANWAVER-UDF and 2DANNULAR-UDF) coupled with two-phase flow models under the Eulerian–Eulerian framework, to predict the streamwise liquid film evolution at different spatial and temporal scales. This will be followed by three-dimensional (3D) liquid film CFD modelling using a user-defined subroutine (3DANNULAR-UDF) coupled with the two-phase flow solver to simulate a full 3D annular flow liquid film dynamic behaviour and its propagation along the flow path. The computational results will be validated by comparing with available experimental data of Schubring and Shedd [9–12], and further assessed by the correlations of GrÖnnerud [35] and Chisholm [36] for the accu-

racy of CFD predictions. The findings will be drawn together with discussions, particularly on its implications for efficient heat exchanger design.

## 2. Numerical Method

### 2.1. Governing Equations of Two-Phase Flow Using the Eulerian–Eulerian Approach

In present study, the two-phase flow simulation was performed by solving the mass, momentum and energy conservation Equations for incompressible fluids using a finite volume (FV) method, coupled with two-phase flow sub-models under the Eulerian–Eulerian framework (see, e.g., ANSYS FLUENT theory guide [37]; Anderson and Jackson [38]; Bowen [39]), as described below.

The mass conservation Equation is written as:

$$\frac{\partial}{\partial t}\left(a_q\rho_q\right) + \nabla \cdot \left(a_q\rho_q \vec{v}_q\right) = \sum_{p=1}^{n}\left(\dot{m}_{pq} - \dot{m}_{qp}\right) + S_q \tag{1}$$

where $\vec{v}_q$ is the velocity of phase $q$, with subscripts denoting the mass transfer from $p^{th}$ to $q^{th}$ phases, and 'dotted' $\dot{m}$ represents the mass transfer from the phase '$q$' to the phase '$p$'. The '$n$' indicates the number of '$p$' phase. In the simulation, '$q$' represents the primary phase of the gas and '$p$' presents the secondary phase of liquid, respectively. $S_q$ is the source term of '$q$' phase. $a_q$ is the volume of the fraction of the '$q$' phase. $\rho_q$ is the density of the '$q$' phase.

The momentum conservation Equation considers the momentum balance of the '$q$' phase that represents the liquid secondary phase in the simulation as follows:

$$\frac{\partial}{\partial t}\left(a_q\rho_q \vec{v}_q\right) + \nabla \cdot \left(a_q\rho_q \vec{v}_q \vec{v}_q\right) = -a_q\nabla p + \nabla \cdot \overline{\overline{\tau}}_q + a_q\rho_q \vec{g} + \sum_{p=1}^{n}\left(\vec{R}_{pq} + \dot{m}_{pq}\vec{v}_{pq} - \dot{m}_{qp}\vec{v}_{qp}\right)$$
$$+ \left(\vec{F}_q + \vec{F}_{lift,q} + \vec{F}_{vm,q}\right) \tag{2}$$

where $\overline{\overline{\tau}}_q$ is the stress–strain tensor of the $q^{th}$ phase, $\lambda_q$ and $\mu_q$ are the shear and the bulk viscosity of the phase '$q$', $F_q$ is an external body force, $F_{lift,q}$ is the lift force, and $F_{vm,q}$ is the virtual mass force that is exerted on the particle due to the inertia of the primary mass encountered by the accelerating particle if there are droplets entrained continuously between the '$q$' (gas) and '$p$' (liquid) phases, $R_{pq}$ is an interaction force between the gas and liquid phases, and '$p$' is the pressure for both the gas and liquid phases. $\vec{v}_{pq}$ is the inter-phase velocity, and it is dependent on the mass flow rate. For $\dot{m}_{pq} > 0$, $\vec{v}_{pq} = \vec{v}_p$ implies the mass in phase '$p$' being fully transferred to phase '$q$', and for $\dot{m}_{pq} < 0$, $\vec{v}_{pq} = \vec{v}_q$ indicates the mass in phase '$q$' being fully transferred to phase '$p$', respectively. $\rho$ is the density of gas or liquid phase.

The energy conservation Equation is written as:

$$\frac{\partial}{\partial t}\left(a_q\rho_q h_q\right) + \nabla \cdot \left(a_q\rho_q \vec{u}_q h_q\right) = a_q\frac{\partial p_q}{\partial t} + \overline{\overline{\tau}}_q \cdot \nabla \vec{u}_q - \nabla \cdot \vec{q}_q + S_q + \sum_{p=1}^{n}\left(Q_{pq} + \dot{m}_{pq}h_{pq} - \dot{m}_{qp}h_{qp}\right) \tag{3}$$

where $h_q$ represents the specific enthalpy of the $q^{th}$ phase, $S_q$ is the source term contributed by the sources of enthalpy, $Q_{pq}$ is the intensity of heat exchange between the $p^{th}$ and $q^{th}$ phases, and $h_{pq}$ is the inter-phase enthalpy. $u_q$ is the velocity magnitude of the '$q$' phase in the flow direction. $q_q$ is the heat flux of the '$q$' phase, respectively.

The concentration of gas and liquid phases in two-phase flow is presented on the same meshes (i.e., co-allocated) using either vapour or liquid volume of fraction (see, e.g., Anderson and Jackson [38]; Bowen [39]) as

$$V_q = \int_V a_q dV \tag{4}$$

The proportion of gas and liquid for all phases is depicted by the following Equation and the sum must be equal to unity, i.e.,

$$\sum_{q=1}^{n} a_q = 1 \tag{5}$$

where $V_q$ is the volume of the phase '$q$' in a cell or a domain, $dV$ is the volume of the cell and $V$ is the volume of the domain.

The momentum interfacial exchange between the gas and the liquid continuous phases is accounted by fluid–fluid exchange coefficients using the Schiller and Naumann model [40]. A re-normalisation group (*RNG*) two-Equation $k$–$\varepsilon$ model (denoted as *RNG k-$\varepsilon$* thereafter) (see, e.g. Choudhury et al. [41]; Orszag et al. [42]) derived from the Navier–Stokes Equations is applied to account for the modelling effect of the turbulence kinetic energy, particularly related to the two-phase flow patterns and their transition between the wave and the full annular flows.

$$\frac{\partial}{\partial t}(\rho k) + \frac{\partial}{\partial x_i}(\rho k u_i) = \frac{\partial}{\partial x_j}\left(\alpha_k \mu_{eff} \frac{\partial k}{\partial x_j}\right) + G_k + G_b - \rho\varepsilon + Y_M + S_k \tag{6}$$

$$\frac{\partial}{\partial t}(\rho\varepsilon) + \frac{\partial}{\partial x_i}(\rho\varepsilon u_i) = \frac{\partial}{\partial x_j}\left(\alpha_\varepsilon \mu_{eff} \frac{\partial \varepsilon}{\partial x_j}\right) + C_{1\varepsilon}\frac{\varepsilon}{k}(G_k + C_{3\varepsilon}G_b) - C_{2\varepsilon}\rho\frac{\varepsilon^2}{k} - R_\varepsilon + S_\varepsilon \tag{7}$$

where $G_k$ represents the generation of turbulence kinetic energy due to the mean velocity gradient. $G_b$ is the generation of turbulence kinetic energy due to the buoyancy. The parameters $\alpha_k$ and $\alpha_\varepsilon$ are the inverse effective Prandtl numbers for $k$ and $\varepsilon$, respectively. $S_k$ and $S_\varepsilon$ are the user-defined source terms. $\mu_{eff}$ is the effective viscosity. $Y_M$ is for the fluctuating dilatation in compressible turbulence to the overall dissipation rate. $C_{1\varepsilon} = 1.44$; $C_{2\varepsilon} = 1.92$ as default values. $k$ is the kinetic energy and $\varepsilon$ is the eddy dissipation. $\rho$ is the density and $u$ is the velocity.

### 2.2. Problem Definition

The transient features in a two-phase flow can produce a range of flow patterns from bubbly flow, slug flow, wavy flow, and finally to full annular flow. This process will have great influences on the subsequent flow behaviour even in the downstream region near the pipe exit, and thus it can effectively influence the flow entrainment and mixing of a complete device configuration. This is partly due to a slow mass exchange process occurring between the liquid film and the gas or vapour core [43]. A typical two-phase flow wave pattern can be predicted using the wavy liquid film developed along a wetted wall surface with a gas core flowing through the centre of the pipe, accompanied by liquid droplets entrainment, as illustrated in Figure 1 (see Yao et al. [44] for further details).

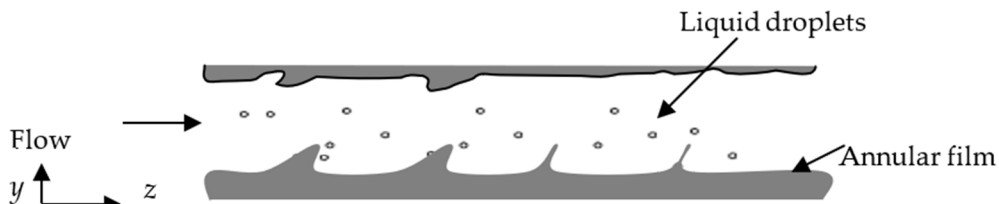

**Figure 1.** A schematic view of annular flow patterns in a cross-section of a horizontal pipe.

In Figure 1, the annular flow pattern shows that the liquid film is usually thicker on the bottom portion of the pipe than that on the top portion, mainly due to the influence of the gravitation force. Sometimes, the over inclusion of the gravitational effect in computational processing can lead to a numerical 'dry-out' phenomenon in the upper region of the pipe walls, where no liquid film coverage can be found. This will result in an under-prediction of

annular flow performance during the transient flow process [45]. Clearly, from the design point of view, this 'dry out' of the liquid film will have a significant impacts on the heat transfer performance of the device. To address this, the CFD simulation results presented in this paper are mainly focused on the flow behaviour of the wavy and the full annular flow regimes, especially on the prediction of the top pipe wall's surface wetness. This is achieved by developing a mathematical model description to depict wave evolution and liquid film thickness distribution circumferentially, to determine the liquid film thickness at the top wall surface by changing the gas and liquid superficial velocity ratio. Thus a well-verified two-phase annular flow model could be further used by industry designers for heat exchanger design practice to improve HVAC in terms of air and water heat pump thermal performance and heat transfer efficiency.

### 2.3. Mathematical Model for Predicting Liquid Annular Film Distributions

The hydrodynamics of the annular flow in a horizontal circular pipe which causes the onset entrainment of liquid or gas components in the continuous phases is likely linked to the deformation of gas and liquid flows at their interfaces, generating the disturbance of wave as a result of entrainment process. The shape of this wave can strongly influence the force acting on the surface of gas and liquid, where the liquid surface interacts with the gas core similar to a flow over a rough wall. In a scenario when shear stress becomes higher than surface tension at the gas and liquid interface, there will be four possible entrainment mechanisms in a horizontal pipe, i.e., (1) roll wave; (2) wave undercut; (3) bubble burst; and (4) liquid, previously identified by Ishii and Grolmes [46]. Butterworth [7] also proposed a spread wave mechanism by investigating interfacial waves at the bottom of the pipe which deformed along the total extension axis, where the pressure of the central gas stream generates forces along the circumferential direction and opposes the gravity effect. All these wave mechanisms can be attributed to the asymmetric liquid film thickness distribution via turbulent flow motions, together with the secondary flow, to influence the wetting of the top wall region in a horizontal pipe. Flores et al. [47] investigated the secondary flow recirculation at high gas and liquid superficial velocity in the horizontal annular two-phase flow regime. The action of the secondary flow was found to promote the cross-section mixing of a two-phase flow by inducing the centrifugal effect under the gravitation field to increase the attribution of the circumferential distribution of interfacial roughness.

Figure 2 gives a schematic representation of a typical 'roll wave' mechanism to spread the wave horizontally along the liquid film. The pattern was adopted to develop the user-defined function (UDF) subroutines (using acronyms as 2DANWAVER-UDF, 2DANNULAR-UDF and 3DANNULAR-UDF thereafter) for modelling 2D wave, 2D and 3D annular flows, respectively. These UDF subroutines was coupled with a commercial CFD solver FLUENT under the Eulerian–Eulerian framework to simulate the wave propagation along the base film with the change in the wave structure and the liquid film thickness in a transient manner.

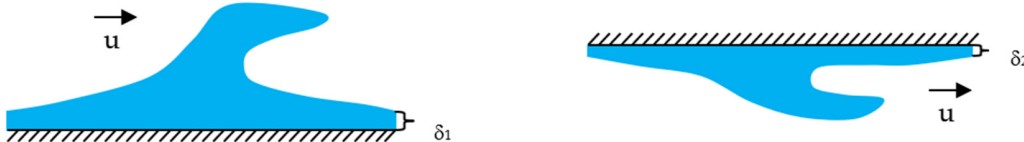

**Figure 2.** A schematic representation of liquid 'roll wave' and film thickness in a horizontal circular pipe: the bottom wave propagation (**left**) and up wave propagation (**right**). Parameter $\delta$ represents the liquid film thickness and $u$ is flow velocity.

The conditions of the initial wave shape and the film thickness can be predefined at the inlet of the pipe, where the maximum base film thicknesses at the bottom and the top walls were set to be 0.1938 mm and 0.125 mm, respectively. The simulation attempts to investigate the influence of the spreading wave on the distribution of liquid film, especially around the upper region of the pipe. The base liquid film thickness can be determined at

the top and the bottom pipe walls in the flow direction for 2D simulation and also in the circumferential direction for 3D simulation, respectively.

Berna et al. [48] described a salient spatial feature of large amplitude travelling waves of different shapes and sizes along the interface between the gas and liquid film, as seen in Figure 3. The wave height is measured by a distance between the wave crest (peak) and the pipe wall, and the film thickness is measured as a distance between half a way of amplitude between the wave crest and the pipe wall. The base film thickness is determined from the pipe wall to the baseline of wave amplitude. The wave penetrating behaviour at the air and the water interface depends on the pressure and velocity gradient generated between the wave crest and the base film at different gas and liquid slip ratios that can affect the equilibrium between the gas and liquid phases.

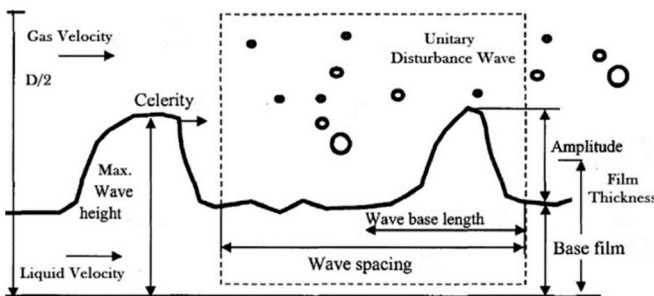

**Figure 3.** A schematic representation of spatial feature of large amplitude travelling waves adopted from Berna et al. [48].

By adopting the wave concept suggested by Berna et al. [48], a mathematical model was established to prescribe the base film and interfacial wave instability features around the pipe wall (see Figure 4). This model considered the base film thickness and the level of instabilities contributing to the wave shape evolution within a complete wave period, using a reciprocal function associated with the time marching in a spatial Cartesian coordinate system.

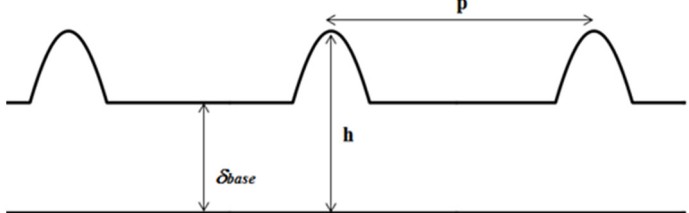

**Figure 4.** A mathematical model describes the base film and the interfacial wave features at the pipe wall. $\delta_{base}$ is the base film thickness; $h$ is the initial height of the wave and $p$ depicts the wave spacing between the wave celerity.

Figure 5 gives a general diagram in which a set of general forms of Equations including UDFs in a transient mode are illustrated. Notations in Figure 5 are as follows, $t$ is time, $\phi(d)$ is the liquid film propagation distance with the time in the Cartesian coordinate system. $S_r$ is a source term related to the base film at the pipe inlet. $r$ is the radius function of time, $\psi(r_\theta)$ is a wave function associated with the wave amplitude envelope. $\theta$ is the rotational angle of the radius. $I$ is a value related to wave amplitude, $x$, $y$ and $z$ represent the Cartesian coordinate system. $R$ is the pipe radius and $\alpha$ is the volume of fraction of the liquid phase, $\delta$ is the film thickness. $d_i$ is the pipe diameter and $h$ is the distance from the wave crest to the pipe wall, respectively.

$$\phi(d) = \phi(x, y, z, t); \ \psi(r_\theta) = \psi(tI \sin \theta) + S_r \tag{8}$$

$$\delta = R - \phi(d) \quad \phi(d) \succ \psi(r_\theta) \quad 0 \prec \alpha \prec 1 \tag{9}$$

$$\delta = R - \psi(r_\theta) \quad \phi(d) \leq \psi(r_\theta) \quad \alpha = 0 \tag{10}$$

$$I = h - \delta \tag{11}$$

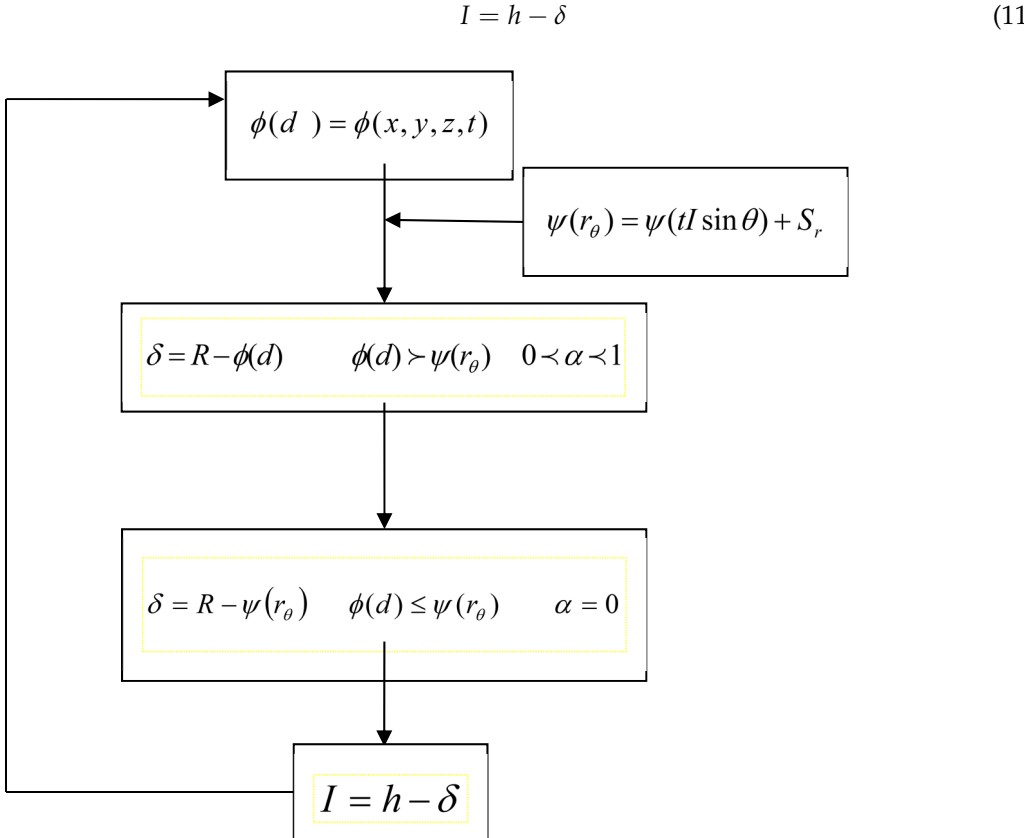

**Figure 5.** A general diagram of equations and UDF used in simulation.

Under the Eulerian–Eulerian framework, the UDF uses the wave function value to control the liquid film development in the 3D Cartesian coordinate system with time marching. If the liquid film volume of the fraction (VOF) value is in the range of 0 to 1, a base film flow thickness will be calculated. Alternatively, a liquid film wave profile will be developed when the VOF value equals to 0.

## 3. Flow Problem and CFD Settings

### 3.1. Computational Domain and Mesh

Figure 6 depicts a schematic view of the computational domain, illustrating a 2D cross-section through the centre line of a straight horizontal 3D circular pipe. The pipe is 1 m in length and 8.8 mm in diameter, the same as that used in the experiments of Schubring and Shedd [9–12].

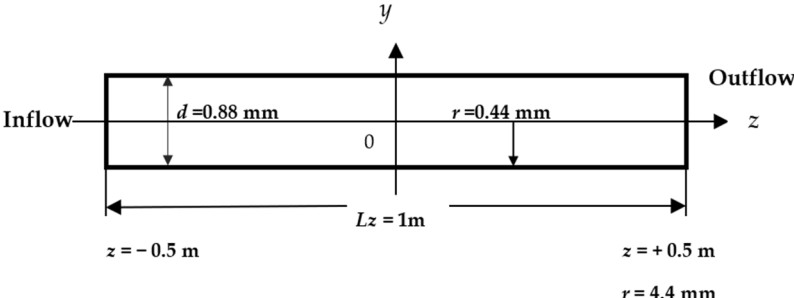

**Figure 6.** A schematic view of 2D computational domain of the *y* and *z* plane.

The grid convergence study was carried out considering a 2D computational domain. It was found that a grid of $100 \times 88$ points is required to achieve numerical solutions which are independent of grid points. Figure 7 displays a Cartesian structured mesh, uniformly

distributed in the streamwise ($z$) direction and stretched (with a stretch factor of 6) in the wall normal ($y$) direction, respectively. The maximum skewness factor was determined at $2.4 \times 10^{-4}$, confirming a good mesh quality obtained. The air flow was of turbulent status as the flow Reynolds number 6466 for the wavy annular flow and 17,587 for the full annular flow, respectively. However, the annular liquid film was of laminar status as the estimated Reynolds number was below 2022. In this respect, the near-wall grid resolution was fine-tuned to satisfy the requirement of chosen SST $k$-$\omega$ turbulence model.

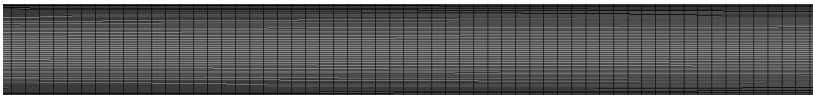

**Figure 7.** A 2D Cartesian structured mesh of $100 \times 88$ points (Yao et al. [44]).

Figure 8 represents the computational meshes of a 3D horizontal pipe along the streamwise $z$-direction and it also displays mid-plane mesh, cross-section mesh and near-wall mesh, respectively. A sweep meshing method was employed to produce a block structured mesh by placing the finer grids in the near wall region and relatively coarser grids at the central region of the pipe. The minimum orthogonal quality was around 0.771 which indicates the mesh quality was good. The flow condition applied for 3D simulation was taken from the experiments, same as those for 2D simulations.

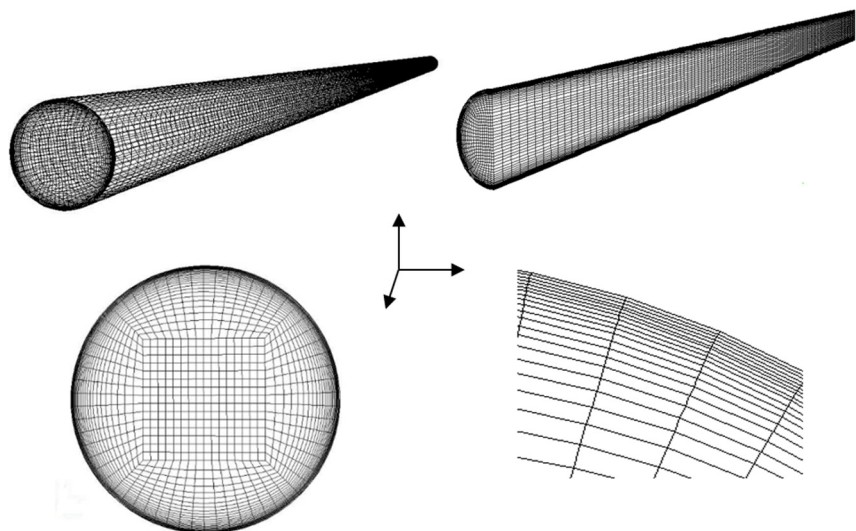

**Figure 8.** 3D computational meshes of a full and half a pipe (**top**), and cross-section views at a mid-plane and in the near-wall (**bottom**).

The 2D grid refinement studies were carried out on three successive meshes of 88,000, 351,000 and 415,000 points with results shown in Figure 9a, where the mean velocity ($U$) was normalised by the bulk velocity at the centre line ($U_o$). The velocity exhibited near linear changes, whilst away from the centre line and then after about $y/R = 0.1$, it followed a parabolic (nonlinear) curve until the pipe wall. The results also showed some velocity variations between locations around $y/R = 0.1$ (i.e., junction between linear and nonlinear), and after $y/R = 0.4$ until the pipe wall ($y/R = 1.0$), the predicted velocities from three meshes were generally in good agreement, indicating a mesh of 88,000 points would be sufficient to achieve a converged solution.

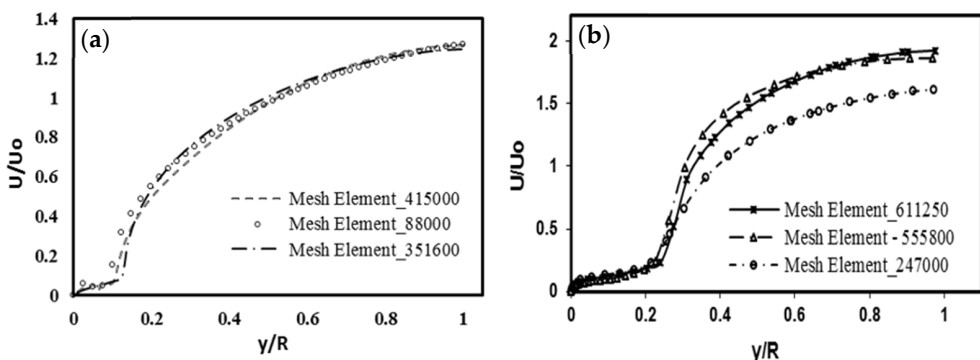

**Figure 9.** Mesh convergence study: (**a**) 2D simulation; (**b**) 3D simulation.

Figure 9b gives 3D mesh convergence studies performed on three successive meshes of 247,100, 555,800 and 611,250 points. The mean velocity was also normalised by the bulk velocity at the centre line. Similar to 2D, the velocity exhibited a linear region from the centre line until $y/R = 0.25$ (i.e., slightly longer than that of 2D) and after this point, it followed a parabolic curve until the pipe wall. It is clear that for a coarse mesh of 247,000, the results showed large deviations compared to that of other two finer meshes of 555,800 and 611,250 grid points. This is mainly due to the near wall mesh density of the two finer meshes being twice as high as the coarse mesh. Based on these studies, it was decided to use a mesh of 611,250 points for 3D simulations.

Whilst both 2D and 3D simulations exhibited similar trends in general, there were noticeable differences in terms of turning point (i.e., around y/R = 0.1 for 2D, and y/R = 0.25 for 3D, respectively) and normalised velocity ratio u/Uo at y/R = 1 (i.e., about 1.2 for 2D and 1.85 for 3D, respectively). The possible cause could be that in the 2D case, the wall shear that caused a friction and pressure drop along the main stream of the gas flow which would be different compared to the annular flow in a 3D circular pipe. In 3D scenarios, the liquid film drainage at the top of the pipe was dominant, due to gravitation effect.

*3.2. Boundary Condition and Simulation Setup*

The inlet gas and liquid superficial velocities and the mass flux were calculated by using the following Equations (see, Schubring and Shedd [12]):

$$U_{sg} = \frac{Gx}{\rho_g} \tag{12}$$

$$U_{sl} = \frac{G(1-x)}{\rho_l} \tag{13}$$

$$G = \frac{\dot{m}_g + \dot{m}_l}{A} \tag{14}$$

$$x = \frac{\dot{m}_g}{\dot{m}_g + \dot{m}_l} \tag{15}$$

where '$G$' is the mass flux, '$x$' is the flow quality, '$\dot{m}$' represents the mass flowrate, '$A$' is the pipe cross-section area, the subscripts '$sg$' and '$sl$' represent superficial gas and superficial liquid, the subscripts '$g$' and '$l$' stand for the gas and liquid phases, respectively.

The inlet air and liquid (water) superficial velocities were calculated using the Equations above and were set at 11.04 m s$^{-1}$ and 0.301 m s$^{-1}$ for 2D wavy annular flow simulations and 33.46 m s$^{-1}$ and 4.25 m s$^{-1}$ for both 2D and 3D full annular flow simulations, respectively, and the fluid properties were taken at ambient temperature of 15 °C. The averaged relative static pressure at the outlet was specified at 0 Pa, and the wall was treated as adiabatic and non-slip. The initial volume of the fraction of liquid (water) at the inlet was

set at 0.104 intermittently for both 2D and 3D simulations, according to the experiments (see, Schubring and Shedd [9–12]).

All simulations were performed in a transient (unsteady) mode to account for the dynamic flow mixing and entrainment processes in the pipe. The buoyancy force effect on the film thickness distributions was considered in the UDF subroutine (ANWAVER-UDF) coupled with the CFD flow solver, in which a second-order phase-coupled numerical scheme was adopted to correlate the pressure and the velocity, implicitly. The simulation also adopted an *RNG k–ε* turbulence model [42]. The interface was modelled using a high-resolution interface capture (HRIC) scheme that consisted of a nonlinear blending of upwind and downwind differencing to enhance the resolution at the air and the water interface [49]. The unsteady flow was then solved by time marching with a fixed time step of $10^{-7}$ s to resolve different spatial and temporal scales of the flow and between adjacent time steps, a total of 20 iterations were employed in the inner loop to ensure the residuals to fall below the prescribed target value before marching to the next step. The SIMPLE algorithm employed for pressure correction iteration was used for coupling the velocity and the pressure fields. The convergence of the solutions was verified for the water VOF by introducing three monitor points in the radial direction of the outlet plane. The *RNG k-ε* turbulence transport Equations were solved using the second-order upwind scheme [50] to give more accurate predictions of turbulence kinetic energy and turbulence dissipation rate. In this approach, the quantities were computed using a multi-dimensional linear reconstruction approach through Taylor series expansion [50].

## 4. Results and Discussion

### 4.1. The Liquid Film Distribution Characterisation

2D CFD simulation was carried out by using UDF subroutines (2DANWAVER-UDF & 2DANNULAR-UDF). The simulation results revealed an instantaneous behaviour of bubbly flow transition to wavy annular flow and finally full annular flow regime in a pipe whilst increasing the air and water superficial velocity ratios in the streamwise flow direction. A wavy annular flow was observed at low air and water superficial velocity ratios, whilst a full annular flow scenario occurred at high superficial velocity ratios, respectively.

Figure 10 illustrates the predicted time-varying two-phase flow patterns (after initial six throughflow) at time instance of 0.434 s, 0.705 s, 0.865 s, 1.320 s, 1.792 s and 2.036 s, respectively. The transient flow pattern can be seen clearly from a chaotic to a wavy annular at a time of 0.705 s and a wavy annular pattern was observed at 0.865 s when a strong wave penetration through the gas and liquid interface was captured, especially around the bottom of the pipe wall. The development of full annular flow feature is started at a time of 1.792 s and completed at a time of 2.036 s.

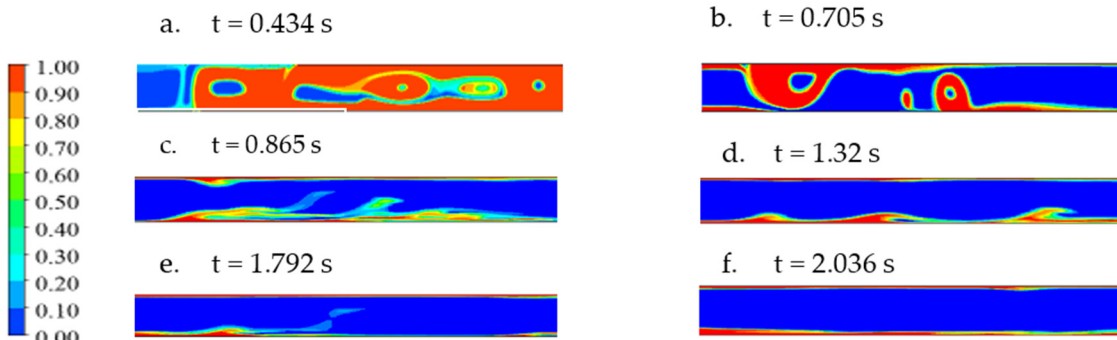

**Figure 10.** Contours of water volume of fraction (VOF) distribution illustrating flow pattern changes from slug flow to full annular flow: (**a**) bubbly/slug flow; (**b**) slug flow; (**c–e**) wavy annular flows; (**f**) full annular flow, respectively. The 'red' colour represents the 'liquid (water) film' and the 'blue' colour represents the 'air'. Other colours represent a mixture of air/water.

The asymmetrical wavy flow pattern around the pipe top/bottom walls (see Figure 10c,d) may be attributed to the buoyancy induced flow instability, combined with the strong interfacial shear stress at the air and water interfaces. The water wave gradually became smoother (see Figure 10e) due to the increase in the ratio of air and water superficial velocity. The water VOF distribution indicated that the air was mainly at the central (core) part of the pipe. The entrainment of air into water (liquid) regime, as depicted by the air 'pockets' at the front of the water slug, was predicted at a time of 0.705 s, consistent with the observed bubbly flow transition to slug flow, followed by the development of wavy annular and full annular flows characterised by the water film coated around the top and bottom pipe walls (see Figure 10c–f). The tendency of the air and water two-phase flow pattern transition predicted by current study is in qualitatively good agreement with the findings of another similar study [51].

Figure 11a depicts velocity vectors of a typical water wave penetrating into the mainstream of air flow. A large scale of water wave was observed at the bottom of the pipe wall and it travelled at a velocity of about 1 m s$^{-1}$, where fluid was moving along the air and water interface. The pattern of water (liquid) phase appeared to be more distorted in the region close to the wall, mainly due to high level of local viscous wall shear stress. This large-scale water wave might also be associated to the weakening of local kinetic energy of the air flow (see 'blue' colour in Figure 11b), in the vicinity of the air and water interface where the air has been entrained into the water (liquid) film to form a clear air and water mixing regime. On the contrary, a small-scale water wave was observed in the top wall region, and it is likely due to the high local kinetic energy of the air flow.

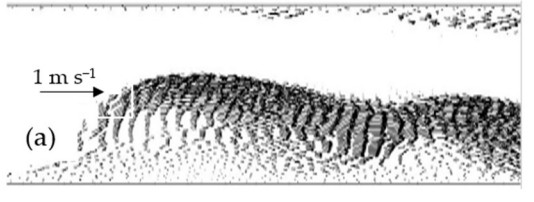 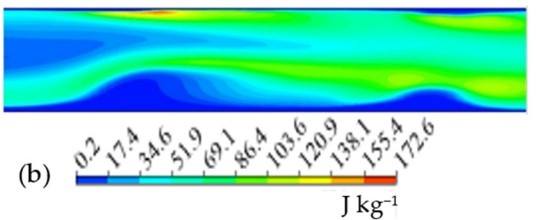

**Figure 11.** An example of instantaneous snapshot of the air and water two-phase flow development displayed at water wavy flow velocity of 1 m s$^{-1}$: (**a**) velocity vectors of water fluid; (**b**) contours of turbulence kinetic energy of air flow.

Figure 12 gives contours of turbulence eddy dissipation rate and it shows higher dissipation rate at the center region of the pipe, surrounded by the distorted air and water interfaces. Those transparent regions near the top and the bottom walls showed a low dissipation rate indicating that there was a predominant liquid film with the water wave penetrating through the air and water interfaces. The water film distributions along the top wall were found to be more uniform and straighter than that along the bottom wall (as depicted in Figure 12a,b, respectively). The higher dissipation rate also implied strong interfacial motions at the air and water interfaces, associated with the high level of turbulence kinetic energy in the same region. A transition of the wavy annular to full annular flow occurred (see Figure 12c).

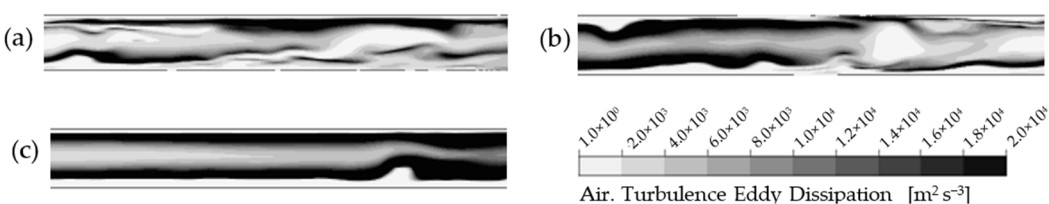

**Figure 12.** Contours of the air flow turbulence eddy dissipation rate illustrating changes from (**a**) and (**b**): wavy annular flow to (**c**) fully developed annular flow.

The characteristics of water (liquid) distributions around the pipe wall were further studied by full 3D simulation with a UDF subroutine (3DANNULAR-UDF), coupled with

a CFD flow solver within the Eulerian–Eulerian framework. Figure 13 shows a prediction of a typical example of a full coverage of water (liquid) film distribution throughout the pipe inner wall surface. The inlet water film distribution was uniform initially and after flow developments, an asymmetrical feature of water film distribution could be seen downstream. The water (liquid) film thickness also saw a trend of a gradual increase around the low half of the pipe wall, whilst there was little change in the upper half of the pipe.

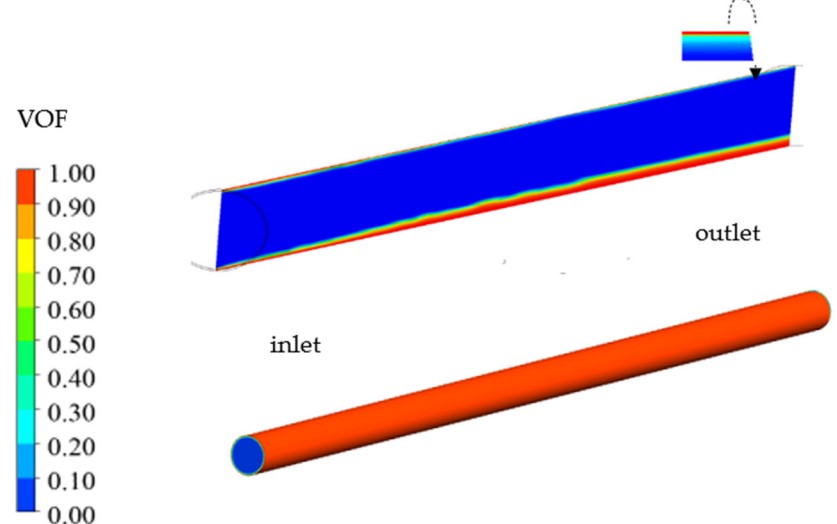

**Figure 13.** Contours of the liquid film distributions on the pipe walls represented by the water volume of fraction (VOF). **Top-right**: a vertical 2D mid-plane through the center of a horizontal pipe (**top-right**); **Bottom-right**: a full 3D liquid film distribution along the pipe wall.

Figure 14 gives the CFD predicted liquid film development at six successive cross planes from the inlet to the outlet. It can be seen from Figure 14b that due to the buoyancy effect, the water (liquid) film around the upper part of the pipe wall started to drain with a larger wavy pattern formed. This trend was further enhanced downstream (see Figure 14c,d) where the phenomena of liquid drainage in the circumferential direction of the pipe wall was clearly observed and this produced a pool of reservoir towards the bottom of the pipe wall, whilst a thin water (liquid) film was developed around the upper pipe wall. A more symmetrical liquid film distribution was predicted near the right and the left sides of the wall surfaces, due to the nature of pipe geometrical configuration, orientation and gravitation force effects (see Figure 14e,f).

### 4.2. The Dynamic Performance of Water Film

Figure 15 illustrates a comparison of 2D and 3D CFD predictions with the experimental date [9]. The averaged base film thickness was calculated using three discrete base film thickness measurements at the two side points in a cross-section plane of the pipe (i.e., a horizontal mid-plane intersecting with the pipe, and here a symmetry was assumed) and one point at the top and one point at the bottom pipe walls, respectively [9].

$$\delta = \frac{\delta_b + 2\delta_s + \delta_t}{4} \tag{16}$$

where $s$, $b$ and $t$ represent the side, the bottom and the top pipe walls of the pipe, respectively. The parameter $\delta$ depicts the base film thickness measured by the distance between the pipe walls to a position just below the wave baseline in the circumferential direction.

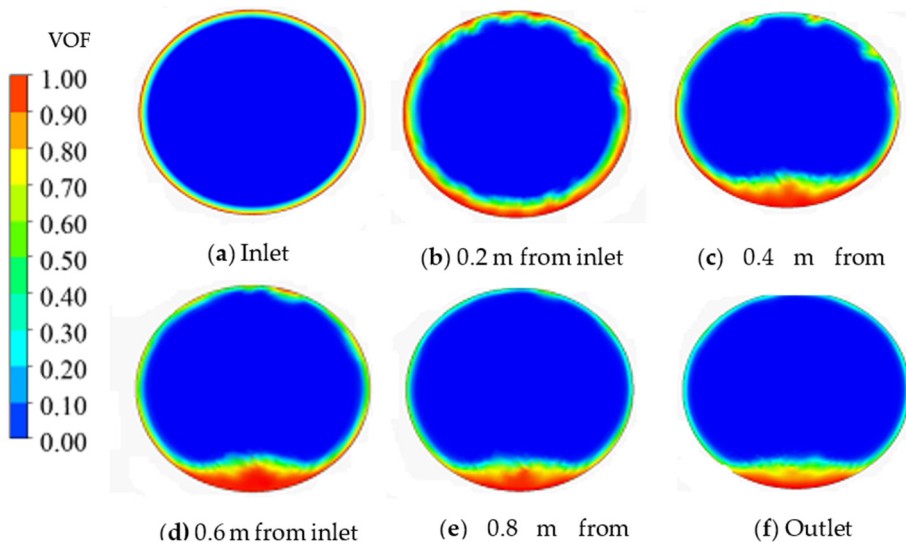

**Figure 14.** Contours of the liquid film distributions at six successive cross-sections from the inlet to the outlet, represented by the water volume of fraction (VOF). (**a**) Inlet; (**b**) 0.2 m away from inlet; (**c**) 0.4 m away from inlet; (**d**) 0.6 away from inlet; (**e**) 0.8 m away from inlet; (**f**) outlet, respectively.

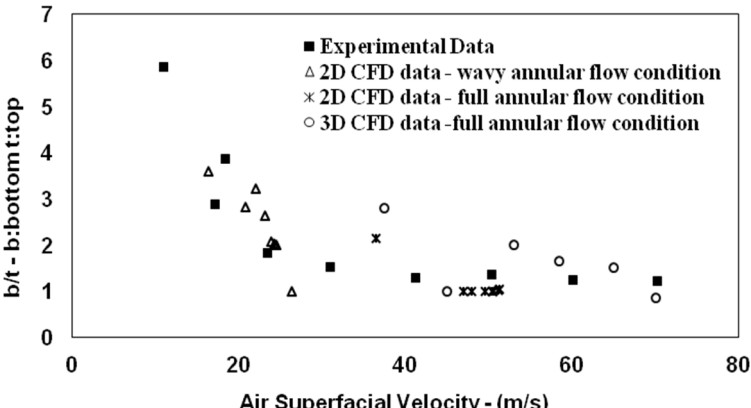

**Figure 15.** CFD predicted air flow superficial velocity variations with the ratio of bottom/top (*b/t*) film thickness, compared with the experimental data [9].

The liquid (water) film thickness ratio at the bottom/top walls (*b/t*) is presented as a function of the air flow superficial velocity. The water base film thickness at the top/bottom pipe wall is determined by the distance from pipe wall to air and water interface, respectively, and it is averaged over the time in the streamwise flow direction. The 2D CFD results with full annular flow condition were in better agreement with the experimental data, especially at high air flow superficial velocity regime than 3D CFD prediction with the same conditions. Both experimental measurements and 2D simulation results showed a trend of symmetrically distributed top and bottom liquid film distributions at a high air flow superficial velocity regime. On the contrary, the 3D CFD prediction exhibited an asymmetrical distribution of the liquid film. This finding is in qualitatively good agreement with that of Abdulkadir et al., who previously observed a similar tendency in their air–water two-phase flow experiment, using a large pipe [52].

The investigation of dynamic performance of water film was carried out to reveal the correlation between the base film thickness and the critical kinetic energy density of the air flow (per cubic meter), for a wavy annular flow transition to a full annular flow it occurred at the kinetic energy density of 600 J m$^{-3}$.

A formula for the kinetic energy calculation can be written as follows:

$$E_{sg} = \frac{1}{2}\rho U_{sg}^2 \tag{17}$$

where $U_{sg}$ is the gas superficial velocity, and $\rho$ is the gas density.

It was found that the liquid (water) film thickness generally decreased with the increase in air flow kinetic energy density. This tendency was captured by all CFD simulations. In particular, the 2D CFD study of a wavy annular flow showed good agreement with the experimental data [12] for low air flow kinetic energy density of 160–300 J m$^{-3}$ (see Figure 16). The discrepancy became more obvious in the base film thickness comparison for kinetic energy density of 1000 J m$^{-3}$ in the full annular flow region. Moreover, the prediction by 3D CFD simulation saw better agreement compared to the experimental data [12] in the full annular flow regime at higher air flow kinetic energy density between 1000 J m$^{-3}$ and 3000 J m$^{-3}$. Compared to the experimental data [12], the discrepancy is likely due to a full buoyancy effect on the liquid (water) film around the circumferential direction of a horizontal pipe wall in the 3D CFD simulation, whilst in the 2D CFD simulation, the buoyancy effect on both the left/right sides of a horizontal pipe could be largely omitted.

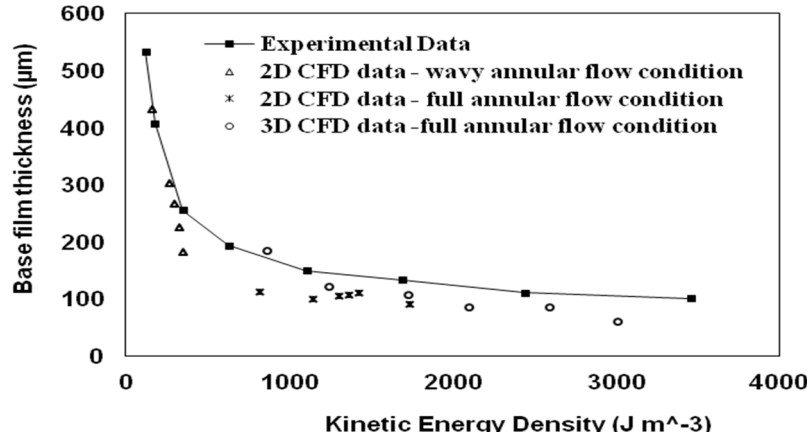

**Figure 16.** CFD predicted air flow kinetic energy density variations with the water base film thickness, in comparison with experimental data [12].

### 4.3. Correlations

GrÖnnerud [35] and Chisholm [36] correlations were applied to assess the accuracy of CFD predictions. In these correlation methods, a wall shear was used by transforming of the correlated interfacial shear based on the estimated pressure drop due to the friction loss on the pipe walls.

#### 4.3.1. GrÖnnerud Correlation

Figure 17 depicts a comparison of wall shear results from both CFD predictions and experimental data [9–11] after data fitting, using an empirical two-phase flow multiplier correlation proposed by GrÖnnerud [35] and later outlined by Ould et al. [53]. The correlated wall shear was calculated using Equation (17), considering the overall force balance along *z*-direction in the horizontal pipe between the shear stress and the pressure gradient throughout the pipe. GrÖnnerud correlation [35] used a further two-phase multiplier to fix the liquid pressure drop, resulting in the friction pressure drop, as given by Equation (18)

$$\tau_{w,corr} = -\frac{d_i}{4}\left(\frac{dp}{dz}\right)_{frict,corr} \tag{18}$$

$$\Delta p_{frict} = \Phi_{gd}\Delta p_L \tag{19}$$

where $\Delta p_L$ is the liquid phase pressure drop and $\Phi_{gd}$ is the two-phase multiplier, calculated using Equations (19) and (20), respectively. The GrÖnnerud two-phase multiplier is associated with the Froude pressure fractional drop and Froude friction factor, presented by Equations (21) and (22), respectively.

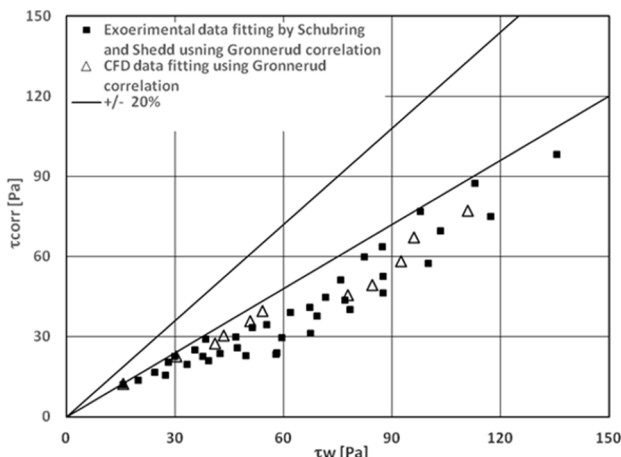

**Figure 17.** Comparison of the data fitting results for present CFD predictions and experimental measurements of Schubring and Shedd [10] using the GrÖnnerud correlation. ($d_i$ = 8.8 mm, $u_{sl}$ > 0.07 m/s).

The two-phase multiplier $\Phi_{gd}$ and $\Delta p_L$ are calculated as:

$$\Delta p_L = 4f_L(L/d_i)\overset{\bullet}{m}{}^2_{total}(1-\chi)^2(1/2\rho_L) \tag{20}$$

$$\Phi_{gd} = 1 + \left(\frac{dp}{dz}\right)_{Fr}\left[\frac{\left(\frac{\rho_L}{\rho_G}\right)}{\left(\frac{\mu_L}{\mu_G}\right)^{0.25}} - 1\right] \tag{21}$$

The Froude pressure frictional drop can be calculated as:

$$\left(\frac{dp}{dz}\right)_{Fr} = f_{Fr}\left[\chi + 4\left(\chi^{1.8} - \chi^{10}f_{Fr}^{0.5}\right)\right] \tag{22}$$

where $\chi$ is the vapour quality in a range of $0 < \chi < 1$ and $f_{Fr}$ is the Froude friction factor associated with the liquid Froude number:

$$f_{Fr} = Fr_L^{0.3} + 0.0055\left(\ln\frac{1}{Fr_L}\right) \tag{23}$$

The Froude number is defined as:

$$Fr = \frac{\overset{\bullet}{m}{}^2_{total}}{gd_i\rho_L^2} \tag{24}$$

where $\overset{\bullet}{m}{}^2_{total}$ is the total mass flow rate, $g$ is the gravitational force, $d_i$ is the pipe inner diameter and $\rho_L$ is the liquid phase density. $f_{Fr} = 1$ for $Fr$ >1 [35,53].

It was found that the GrÖnnerud correlation [35] was consistently underestimating for both CFD results and the experiments of Schubring and Shedd [10] (see Figure 17). The majority of data points were below the –20% error line of the averaged mean correlation. The mean absolute error (MAE) or the root mean square (RMS) error of GrÖnnerud correlation was determined at 42.58% and 44.86%, respectively [10]. The correlated CFD results showed an overall under-prediction with maximum errors within 40%, similar to those from experimental data using the same correlation. The over prediction becomes

particularly predominant in the high wall shear region of greater than 80 Pa, the correlated CFD data were found to produce errors higher than that estimated at low wall shear region. The tendency of CFD predictions were consistent with those given by experimental results. Overall, it can be concluded that the GrÖnnerud correlation cannot produce the error estimation for both present CFD predictions and published experimental data within +/−20% error margin.

### 4.3.2. Chisholm Correlation

Chisholm [36] developed an empirical correlation using the Chisholm parameter and applying Equations (25)–(31) to calculate the two-phase multiplier which was then employed to correct the liquid pressure drop (see Equation (24)). This led to the change in a pressure frictional drop as

$$\left(\frac{dp}{dz}\right)_{fract} = \left(\frac{dp}{dz}\right)_{Lo} \Phi^2 \tag{25}$$

$$\Phi^2_{L_o} = 1 + \left(Y^2 - 1\right)\left[B\chi^{(2-n)/2}(1-\chi)^{(2-n)/2} + \chi^{2-n}\right] \tag{26}$$

where $B = 4.8$ is a constant for $0 < Y < 9.5$ and $\dot{m}^2_{total} \leq 500$ kg/m$^2$ s. Parameter $n$ is an exponential factor taking a constant value of 0.25.

$$Y^2 = \frac{(dp/dz)_{Go}}{(dp/dz)_{Lo}} \tag{27}$$

where $(dp/dz)_{Go}$ and $(dp/dz)_{Lo}$ are the gas and the liquid frictional pressure gradients associated with the total mass flow rate and flow friction factor, respectively.

$$\left(\frac{dp}{dz}\right)_{Lo} = f_L \frac{2\dot{m}^2_{total}}{d_i\rho_L} \tag{28}$$

$$\left(\frac{dp}{dz}\right)_{Go} = f_L \frac{2\dot{m}^2_{total}}{d_i\rho_G} \tag{29}$$

Parameters $f_L$ and $f_G$ are the friction factor of the liquid and the gas. For a laminar flow, it is:

$$f = \frac{16}{\text{Re}} \tag{30}$$

For turbulent flow, it is:

$$f = \frac{0.079}{\text{Re}^{0.25}} \tag{31}$$

The parameter Re is the Reynolds number calculated by:

$$\text{Re} = \frac{\dot{m}_{total}d_i}{\mu} \tag{32}$$

where $\mu$ is the dynamic viscosity.

Figure 18 gives the data fitting results using Chisholm correlation. It is clear that the Chisholm multiplier correlation [36] performed better than that of GrÖnnerud [35] for both the present CFD predictions and the experimental data, despite the Chisholm multiplier model [36] marginally under-predicting the air flow velocity magnitude in the low wall shear regime (see Figure 18). The MAE or RMS error of the Chisholm correlation was determined at 25.67% and 36.98%, respectively [10]. Using this multiplier correlation, CFD data were overall fitted well using the data measured by Schubring and Shedd [10]. Compared to −20% error line of the averaged mean correlation of Chisholm two-phase multiplier, the maximum error of 30% was observed, over-predicting the friction wall

shear stress below 45 Pa (see Figure 18) where the experimental data showed a clear trend of overestimation of wall shear stress at low air flow velocity region [10]. In general, the correlated CFD predictions were slightly better than those experimental data in the low wall shear region, as they were almost within the + 20% error line, despite of the underestimation of the friction pressure drop at a wall shear of 30 Pa. The discrepancies between CFD predictions and experimental data could be due to either experimental measurement uncertainty or the accuracy of CFD predictions. Nevertheless, the agreement between the present CFD predictions and the experimental data [10] was largely improved in the high wall shear regime.

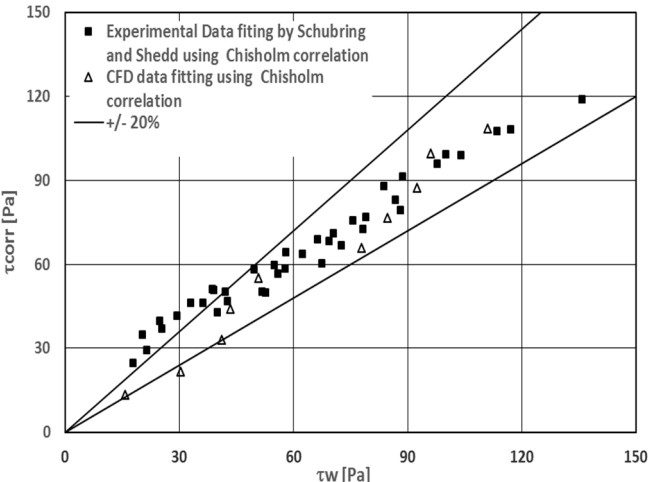

**Figure 18.** Comparison of the data fitting results for present CFD predictions and experimental measurements of Schubring and Shedd [10] using the Chisholm correlation. ($d_i$ = 8.8 mm, $u_{sl}$ > 0.07 m/s).

Overall, the Chisholm multiplier correlation [36] produced better data fitting results for both CFD predictions and the experimental compared to that of the GrÖnnerud [35] correlation.

## 5. Conclusions

Transient CFD simulations were carried out to investigate the characteristics of two-phase flow pattern changes from a wavy annular flow to a full annular flow in a horizontal circular pipe of 8.8 mm diameter. The user-defined function (UDF) subroutines of 2D and 3D annular film models were developed for capturing wavy and full annular flows and they were successfully implemented in a commercial flow solver under the Eulerian–Eulerian framework.

2D CFD simulation successfully replicated the process of a wavy annular flow transition to a full annular flow, and the characteristics of a fully developed annular flow were also captured in the 3D CFD simulation. It was found that the water (liquid) film thickness distributions were in good agreement with available experimental data [9–11]. The air and water turbulence kinetic energy and turbulence eddy dissipation rates were generally higher for a full annular flow than a wavy annular flow. This may contributed to a high water film coverage on the top of pipe wall, due to the propagation of water wave induced droplets. The water film thickness increased at the bottom of pipe wall and decreased at the top of pipe wall, with the increase in the air and water superficial velocities. The performance investigations using two-phase multiplier correlations showed that the GrÖnnerud model [35] gave a poor correlation of wall shear stress between the CFD and experiment. On the other hand, the Chisholm model [36] showed better correlated wall shear stress characteristics between CFD predictions and experimental measurements within maximum 20% errors. Nevertheless, the tendencies of correlated CFD results were overall consistent with the experimental data for the flow region considered in the present study.

**Author Contributions:** Conceptualization, J.Y. and Y.Y.; methodology, J.Y. and Y.Y.; software, J.Y.; validation, J.Y.; formal analysis, J.Y.; investigation, J.Y.; resources, J.Y.; data curation, J.Y.; writing—original draft preparation, J.Y.; writing—review and editing, Y.Y.; visualization, J.Y.; supervision, Y.Y.; project administration, J.Y.; funding acquisition, J.Y. All authors have read and agreed to the published version of the manuscript.

**Funding:** This research was partly funded by Mitsubishi Electric R&D Research Centre Europe B.V. MERCE-UK through contract (E&E-CON1-9).

**Institutional Review Board Statement:** Not applicable.

**Informed Consent Statement:** Not applicable.

**Data Availability Statement:** The data presented in this study are available on request from the corresponding author.

**Acknowledgments:** The first author would like to acknowledge the sponsorship from the Mitsubishi Electric R&D Research Centre Europe B.V. MERCE-UK.

**Conflicts of Interest:** The authors declare no conflict of interest.

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
