# Peer review of "Transient CFD Modelling of Air–Water Two-Phase Annular Flow Characteristics in a Small Horizontal Circular Pipe"

_fluids, doi:10.3390/fluids7060191_

Round 1
Reviewer 1 Report
- Page 1, line 11.....preciously predict........ Use scientific words and this should replicated throughout the manuscript.
- Line 30-33…… Most recently, 30 the renewable system such as air. How is rhis related to the subject title? Reference required.
- Line 41-44, Thus this makes the present re- 41 search necessity …….. This is confusing…Rephrase.
- Page 2, line 52-55… An earlier observation of the liquid film 52 draining circumferentially and the secondary flow influence was made by Pletcher et al. 53 (1965) and further detailed by Lanrinat et al. (1984) and Lin et al. (1986), respectively. How is the use of the word respectively connected to the preceding text?
- Line 71…. The two-phase flow often exhibits strong instantaneous feature primarily due to the instability mechanism. Ref??
In conclusion, the paper is well written with a few grammatical errors however the authors need to clearly and explicitly demonstrate the novelty of this work if any
Author Response
Here are our responses to reviewer’ comments.
- Page 1, line 11.....preciously predict........ Use scientific words and this should replicated throughout the manuscript.
Ans: the word ‘preciously’ has been removed as suggested.
- Line 30-33…… Most recently, 30 the renewable system such as air. How is this related to the subject title? Reference required.
Ans: as suggested, we have included a new reference paper (Pourahmad et al. 2021) in supporting our statement.
- Pourahmad, S.M. Pesteei, H. Ravaeei, and S. Khorasani. 2021. Experimental study of heat transfer and pressure drop analysis of the air/water two-phase flow in a double tube heat exchanger equipped with dual twisted tape turbulator: simultanepous usage of active and passive methods. Journal of Energy Storage 44, part B, 103408 (Online). https://doi.org/10.1016/j.est.2021.103408
- Line 41-44, Thus this makes the present re- 41 search necessity …….. This is confusing…Rephrase.
Ans: we have rephrased these sentences as this reviewer suggested.
- Page 2, line 52-55… An earlier observation of the liquid film 52 draining circumferentially and the secondary flow influence was made by Pletcher et al. 53 (1965) and further detailed by Lanrinat et al. (1984) and Lin et al. (1986), respectively. How is the use of the word respectively connected to the preceding text?
Ans: we have removed the word ‘respectively’ and revised the sentence accordingly to make it clear and readable.
- Line 71…. The two-phase flow often exhibits strong instantaneous feature primarily due to the instability mechanism.
Ans: we have included a reference paper here (O’Neill et al. 2020) in supporting the statement.
O’Neill, L.E., Mudaear, I., 2020. Review of two-phase flow instabilities in macro- and micro-channel system (Review). International Journal of Heat and Mass Transfer 157, 119738. DOI: 10.1016/j.ijheatmasstransfer.2020.119738
- In conclusion, the paper is well written with a few grammatical errors however the authors need to clearly and explicitly demonstrate the novelty of this work if any
Ans: thank you for your positive comments. We have made a thorough revision by correcting grammatical errors and also improve the paper quality by demonstrating the novelty of the work, e.g. the development of in-house UDF and results discussion and analysis.
Reviewer 2 Report
The manuscript studied modeling the air-water two-phase annular flow employing the CFD approach. The results are claimed to be consistent with the existing experimental data. The paper is recommended to be published after the following minor revisions are done:
- The functional procedure of each UDF subroutine should be illustrated for potential readers.
- In fig 5, the value of z should be ranged from -0.5 to +0.5.
- The rationality of the different results in Fig 8 (a) and (b) should be explained.
- What is the definition of m in Eq. 14 & 15?
- The numerical results should be validated and given in the paper before the results and discussion section.
- What is the correlation coefficient or R^2 value of the correlation provided in Fig 16 and Fig 17?
Author Response
We have appreciated reviewer’s comments. We have made following corrections.
Q1: The functional procedure of each UDF subroutine should be illustrated for potential readers.
Reply: We agree in principle with this reviewer for his/her comment on ‘functional procedure of each UDF subroutines to be illustrated for potential readers’. However, this work is fully sponsored by the industry; therefore, we cannot release the UDF subroutine details under the non-disclosure agreement with the company.
Q2: In fig 5, the value of z should be ranged from -0.5 to +0.5.
Reply: Thank you. We have changed the value of z to be -0.5 to +0.5
Q3: The rationality of the different results in Fig 8 (a) and (b) should be explained.
Reply: As suggested, we have added a paragraph to discuss rational between 2D and 3D simulation results.
Q4: What is the definition of m in Eq. 14 & 15?
Reply: It should be mass flow rate . There are issues when placing original manuscript to MDPI template. We have now replaced with correct eq. 14 & 15.
Q5: The numerical results should be validated and given in the paper before the results and discussion section.
Reply: We agree with this reviewer on the importance of validating numerical results before applications. Indeed, we presented thorough mesh convergence studies (see Figures 8a, 8b) to ensure any numerical inaccuracy to be minimised, and finally to compare our simulation results with available experimental data (see Figures 14-17). We also cited a published paper by the present authors to reinforce this point.
Q6: What is the correlation coefficient or R^2 value of the correlation provided in Fig 16 and Fig 17?
Reply: We cannot find the R^2 value as this reviewer mentioned, but have added the values of mean absolute error (MAE) and the root mean square (RMS) error for both GrÖnnerud and Chisholm correlations, after revisiting the original reference papers.
Reviewer 3 Report
I read the manuscript with great interest. Much attention was paid to analyzing the problem and researching the method. CFD simulation is very convenient for creating scenarios and understanding the possible consequences of hypotheses or applying models. The manuscript has been drawn up with great care and is written in a very understandable language and the path described is clear and didactic. I would like to emphasize just a small clarification which, in my opinion, deserves to be made explicit. In line 336, a grid of 100x88 points is assumed for the mesh. The choice is most likely correct, but the motivation is not clear and, if possible and existing, it would be better to link this choice to a bibliographic reference.
Author Response
Q1: In line 336, a grid of 100x88 points is assumed for the mesh. The choice is most likely correct, but the motivation is not clear and, if possible and existing, it would be better to link this choice to a bibliographic reference.
Reply: We appreciate very positive comments made by this reviewer. As suggested, we have added one of our pervious published work (Yao et al. 2016, ref. 53) to address this particular issue raised by this reviewer.
Round 2
Reviewer 1 Report
However, I have now gone through the copy attached to your email, and I can confirm that the authors have taken my remarks into account, resulting in my recommendation for publication.